# Alterations in Retinal Vessel Diameters in Patients with Retinal Vein Occlusion before and after Treatment with Intravitreal Ranibizumab

**DOI:** 10.3390/jpm13020351

**Published:** 2023-02-17

**Authors:** Evaggelia Aissopou, Athanasios Protogerou, Panagiotis Theodossiadis, Petros P. Sfikakis, Irini Chatziralli

**Affiliations:** 12nd Department of Ophthalmology, National and Kapodistrian University of Athens, 12462 Athens, Greece; 2Cardiovascular Prevention Unit, Department of Pathophysiology, National Kapodistrian University of Athens, 11527 Athens, Greece; 3Rheumatology Unit, National Kapodistrian University of Athens, 11527 Athens, Greece

**Keywords:** retinal vein occlusion, retinal vessel diameters, anti-VEGF

## Abstract

**Purpose**: To investigate the alterations of retinal vessel diameters in patients with macular edema secondary to retinal vein occlusion (RVO), before and after treatment with intravitreal ranibizumab. **Methods**: Digital retinal images were obtained from 16 patients and retinal vessel diameters were measured before and three months after treatment with intravitreal ranibizumab with validated software to determine central retinal arteriolar and venular equivalents, as well as arteriolar to venular ratio. **Results**: In 17 eyes of 16 patients with macular edema secondary to RVO (10 with branch RVO and 6 with central RVO) aged 67 ± 10.2 years, we found that diameters of both retinal arterioles and venules were significantly decreased after intravitreal ranibizumab treatment. Specifically, the central retinal arteriolar equivalent was 215.2 ± 11.2 μm at baseline and 201.2 ± 11.1 μm at month 3 after treatment (*p* < 0.001), while the central retinal venular equivalent was 233.8 ± 29.6 μm before treatment versus 207.6 ± 21.7 μm at month 3 after treatment (*p* < 0.001). **Conclusions**: A significant vasoconstriction in both retinal arterioles and venules in patients with RVO was found at month 3 after intravitreal ranibizumab treatment compared to baseline. This could be of clinical importance, since the degree of vasoconstriction might be an early marker of treatment efficacy, compatible with the idea that hypoxia is the major trigger of VEGF in RVO. Further studies should be conducted to confirm our findings.

## 1. Introduction

Retinal vein occlusion (RVO) is the second most common retinal vascular disorder in adults, following diabetic retinopathy, and its incidence increases with age [1,2,3]. Rogers et al. tried for the first time to estimate the prevalence of RVO worldwide and found that 16.4 million people aged 30 years and above were suffering from RVO in 2008 [2]. In 2015, the global prevalence of RVO was found to be about 0.77% in people aged 30–89, affecting about 28 million people worldwide [3]. RVO is a significant cause of visual impairment with cystoid macular edema (CME) and macular ischemia being the most important sight-threatening complications in patients with RVO [4]. Depending on the site of occlusion, RVO can be broadly classified as central retinal vein occlusion (CRVO), branch retinal vein occlusion (BRVO) or hemiretinal vein occlusion (HRVO) [4].

In terms of the pathogenesis of RVO, it is considered to be multifactorial and is mainly based on Virchow’s triad, consisting of hemodynamic changes (venous stasis), vascular endothelial damage and hypercoagulability [5]. Hypertension, dyslipidemia and diabetes mellitus have been reported as primary systemic risk factors, while hematological disorders and abnormalities in the coagulation system may also play a role in the pathophysiology of RVO [5]. In CRVO, arteriosclerosis and hypertension have been found to increase the rigidity of the central retinal artery, which shares a common adventitial sheath with the adjacent central vein, exerting compression on the vein at the level of the lamina cribrosa and increasing intravascular hydrostatic pressure, which may cause vein obstruction [6,7]. In BRVO, the crossing artery over a vein at areas of arterio–venous crossings leads to turbulent blood flow and damage to the endothelium with thrombus formation [8,9].

Interestingly enough, the role of inflammation in the pathogenesis of RVO is also fundamental. The impaired endothelium of the affected vein may activate the release of inflammatory mediators with vascular endothelial growth factor (VEGF) the most prominent, inducing the break-down of the blood–retina barrier and causing macular edema [10]. Specifically, VEGF is a strong promoter of vascular permeability, expressed at high concentrations in the setting of macular edema and released in response to hypoxia. In eyes with RVO, elevated intraocular levels of VEGF induce disruption of the endothelial zonula occludens, fenestrations in endothelial cells, fragmentation of endothelium, and degenerative changes in endothelial basement membranes [11]. Moreover, VEGF, via a nitric oxide (NO) dependent mechanism, acts as strong retinal vascular vasodilator [12]. Maar et al. showed that grid photocoagulation in BRVO led to retinal venous vasoconstriction and therefore a less hypoxic retina, as hypoxia is a potent vasodilator of retinal vessels [13,14]. Therefore, the use of anti-VEGF agents plays an important role in the treatment of macular edema secondary to RVO, since large pivotal trials have proven their efficacy and safety [15].

So far, few studies have examined the effect of therapeutic procedures such as laser photocoagulation [13,16], intravitreal ranibizumab [17,18], bevacizumab [19] and dexamethasone implant [20,21] on retinal vessel calibers in patients with macular edema due to RVO. The majority of previous studies reported that there was vasoconstriction of arterioles and venules after intravitreal anti-VEGF or laser treatment in RVO patients. However, the studies so far had short-term follow-up of 1-month [18], while they included only patients with BRVO [13,16,17,19]. In light of the above, the purpose of this study was to evaluate non-invasively the retinal hemodynamic consequences of intravitreal ranibizumab treatment in RVO, based on the hypothesis that RVO is associated with a vasoconstrictor effect on retinal vessels, providing a follow-up of 3 months.

## 2. Materials and Methods

Participants in this study were 16 patients with treatment-naïve macular edema due to RVO, who were treated with intravitreal ranibizumab at the 2nd Department of Ophthalmology, National and Kapodistrian University of Athens, Greece between 1 October 2020 and 28 February 2022. Inclusion criteria were the following: (1) patients aged older than 18 years and able to provide informed consent; (2) treatment naïve patients with macular edema secondary to RVO (central subfield thickness-CST ≥320 μm); (3) best-corrected visual acuity (BCVA) between 20/200 and 20/25 (Snellen equivalent) in the studied eye. Patients were excluded from the study if they had diabetes mellitus, hypertension or other systemic disease, retinal diseases other than RVO, history of ocular inflammation, uncontrolled glaucoma with intraocular pressure (IOP) ≥30 mmHg, previous laser photocoagulation, previous intravitreal injection of anti-VEGF or steroids, trauma, any ocular surgery within the previous 6 months and significant media opacities (from cornea or lens) that could preclude adequate retinal imaging, affecting the reliability of measurements. The study was in accordance with the tenets of the Declaration of Helsinki and was approved by the institutional review board of Attikon University Hospital (Reference number: 699/2019). Informed consent was obtained from all participants before entering the study.

At baseline and before any treatment, all participants underwent a complete ophthalmic examination, including BCVA measurement by means of Snellen charts, slit lamp biomicroscopy, IOP measurement using Goldmann applanation tonometry, and dilated fundoscopy. The diagnosis of RVO was based on clinical findings, including presence of retinal hemorrhages, retinal vein dilatation, tortuosity, flame-shaped and dot-blot hemorrhages, with or without optic disc hyperemia, while confirmed by retinal imaging. Specifically, all patients underwent infrared fundus photography, spectral-domain optical coherence tomography (SD-OCT) and fluorescein angiography (FA) using Heidelberg Spectralis (Spectralis HRA+OCT, Heidelberg Engineering, Heidelberg, Germany). A quantitative retinal grading was conducted by a well-trained ophthalmologist (EA) blinded to clinical data, including patient’s history, OCT findings and the time-point of examination (before or after treatment). For each photograph, the calibers of the six largest retinal arterioles and venules passing through a zone between 0.5 and 1.0 disc diameters from the optic disc margin were measured and analyzed using a Static Retinal Vessel Analyzer (SVA-T and Vesselmap 2 software [22], Visualis, Imedos Systems UG, Jena, Germany) (Figure 1). All measurements were performed manually, based on the clinical knowledge that (a) retinal arterioles are smaller than venules and (b) retinal venules are darker and more tortuous than arterioles. Then, the calculations were automatically performed by the validated software. The measurements were summarized using formulas described by Knudtson and Hubbard [23] to compute the central retinal arteriolar equivalent (CRAE) and the central retinal venular equivalent (CRVE), representing the average internal caliber of the retinal arterioles and venules, respectively. In addition, CRAE and CRVE were used to estimate the arteriolar to venular ratio (AVR). The intra-observer reproducibility of retinal vascular measurements was excellent, as indicated by the intraclass correlation coefficient (>0.9).

All patients were treated with a loading phase of 3 monthly intravitreal ranibizumab injections and were examined at month 3 after treatment initiation, using infrared fundus photography and SD-OCT. Comparisons between month 3 and baseline were performed for all parameters (CRAE, CRVE, AVR).

Statistical analysis was performed using the SPSS statistical package (IBM Corp., version 21.0, Armonk, NY, USA). The Kolmogorov–Smirnov test and histograms were used to test normality of the variables’ distribution. Normally distributed variables are presented as mean±standard deviation and categorical variables as counts with frequencies. Comparisons of retinal vessel diameters before and after treatment were performed, using independent samples *t*-test. Correlations between variables were performed using Spearman’s test. The level of statistical significance was set at *p* < 0.05.

## 3. Results

Table 1 shows the demographic and clinical characteristics of our study sample at baseline. We included 17 eyes of 16 patients with macular edema secondary to RVO (10 with BRVO and 6 with CRVO). The mean age of the study sample was 67 ± 10.2 years. A total of 9 out of 16 patients were male (56.3%) and 7 female (43.7%). At baseline, the mean IOP was 18.3 ± 2.1 mmHg and the mean BCVA 0.28 ± 0.13 (decimal scale). The mean CST was 427.4 ± 63.7 μm at baseline and significantly decreased to 387.2 ± 45.8 μm at month 3 after intravitreal ranibizumab treatment. The mean CRAE was 215.2 ± 11.2 μm at baseline and at month 3 after treatment 201.2 ± 11.1 μm, being significantly lower (*p* < 0.001). The mean CRVE was 233.8 ± 29.6 μm before treatment and decreased significantly to 207.6 ± 21.7 μm 3 months after ranibizumab initiation (*p* < 0.001). However, no statistical significance was observed as far as AVR was concerned (0.93 ± 0.10 before and 0.98 ± 0.12 after ranibizumab treatment, respectively, *p* = 0.107) (Table 2). It is also worthy to note that no correlation between the reduction in macular edema and CRAE or CRVE difference was found (*p* = 0.082 and *p* = 0.103 for CRAE and CRVE, respectively).

When we studied the ten patients with BRVO separately, we found that the mean CRAE was 215.8 ± 10.3 μm before, and 203.6 ± 10.3 μm after, ranibizumab treatment (*p* = 0.002). The mean CRVE was 221.0 ± 15.6 μm and 197.2 ± 12.2 (*p* < 0.001) before and after ranibizumab treatment, respectively. A borderline statistical significance was found in the mean AVR (0.98 ± 0.07 and 1.04 ± 0.09 before and after treatment, respectively, *p* = 0.043).

Accordingly, regarding the six CRVO patients, the mean CRAE was 214.4 ± 14.2 μm and 197.5 ± 11.9 μm before and after ranibizumab treatment, respectively, (*p* = 0.007), while the mean CRVE was 253.5 ± 35.5 μm and 223.7 ± 23.7 μm before and after treatment, respectively, (*p* = 0.031). No statistical significance was found for mean AVR (0.86 ± 0.10 and 0.89 ± 0.10 before and after treatment, respectively, *p* = 0.434).

## 4. Discussion

In the present study, we investigated the effect of intravitreal ranibizumab treatment on the diameter of retinal vessels in patients with macular edema secondary to RVO. A significant vasoconstriction of both retinal arterioles and venules was found 3 months after treatment with intravitreal ranibizumab compared with baseline. This result remained the same when we analyzed CRVO and BRVO patients separately.

Previous studies have investigated the effect of several treatment modalities on retinal vessel calibers in patients with macular edema due to RVO. Our results are in line with the findings of two studies that used intravitreal ranibizumab in 27 patients with BRVO and 11 eyes with RVO, respectively [17,18]. Specifically, both studies reported that there was vasoconstriction of arterioles and venules after intravitreal ranibizumab treatment in patients with RVO. However, Sacu et al. included only BRVO patients in their 3-month study [17], while Corvi et al. presented short-term results of 1-month after ranibizumab treatment [18]. On the other hand, Nagaoka et. al. found that one injection of bevacizumab had no impact on retinal vessel diameters in their study of 33 eyes with BRVO [19]. It is well-known that VEGF-associated vascular permeability and hemodynamic changes are mediated by increased NO production [24]. Therefore, the mechanism by which anti-VEGF treatment decreases retinal vessel diameter is through the inhibition of NO production. The reduction in VEGF levels and inflammatory mediators with anti-VEGF injection leads to a decrease in retinal edema and might help the blood–retinal barrier to regenerate.

Sectorial or grid photocoagulation in ischemic or edematous areas have beneficial effects in BRVO, especially in cases of severe ischemia that may lead to neovascularization [5]. Arnaarsson et al. and Maar et al., studied the effect of laser treatment in 12 BRVO patients and in 14 BRVO eyes, respectively, and both concluded that laser treatment leads to the vasoconstriction of occluded retinal venules [13,16]. Regarding retinal arterioles, the results of these studies were not similar, as the first found significant vasoconstriction of adjacent retinal arterioles and the second reported vasoconstriction, but not at a statistically significant level [13,16]. Laser treatment in RVO aims to reduce hypoxia [25,26], whereas intravitreal anti-VEGF agents aim to decrease retinal VEGF levels, diminishing the vasodilator tone and edema formation. Furthermore, laser treatment in BRVO induces constriction of the occluded venule and the adjacent retinal arteriole with no impact on the healthy vessel diameters.

Apart from laser photocoagulation and intravitreal anti-VEGF agents, intravitreal steroids have been used in the treatment of macular edema secondary to RVO. Two studies that investigated the effect of an intravitreal injection of a sustained-release intravitreal dexamethasone implant (Ozurdex; Allergan, Irvine, CA, USA) in 17 and 40 eyes with RVO, respectively, found that retinal venular diameters were narrower after treatment, whereas retinal arterioles showed no response [20,21]. Furthermore, Eibenberger et al. reported a significant decrease in retinal venules only in CRVO and not in the studied BRVO population [21], whereas Yilmaz Tugan et al. demonstrated the vasoconstriction of venules in both CRVO and BRVO patients [20], showing controversial results. These findings could be attributed to the fact that intravitreal dexamethasone acts by stabilizing the endothelial cell tight junctions and reducing intraocular cytokine production [7].

The effect of anti-VEGF treatment on retinal vessel diameters has also been studied in other ocular retinal diseases, such as neovascular age-related macular degeneration (AMD) [8,9] and diabetic macular edema [27,28,29,30,31]. Mendrinos et al. [27] concluded that intravitreal ranibizumab injections (median number four) induced a sustained narrowing of the retinal arterioles in ten patients with neovascular AMD after one-year follow-up. Similarly, Fontaine et al. [28] found that three repeated injections of bevacizumab in 23 patients with neovascular AMD led to a decrease in retinal arteriolar diameter. In patients with diabetic macular edema the results were divergent. Two studies concluded that anti-VEGF treatment caused a significant narrowing in the retinal vessels’ diameters [30,32], while one study found a significant decrease in CRAE but not in CRVE [29]. Three studies showed vasoconstriction of retinal vessels, but did not reach statistical significance [31,33,34].

The main limitation of our study is the small number of the included eyes and the short follow-up period. However, we defined strict exclusion criteria, given that patients with systemic diseases were not included in the study. Moreover, no control group was included due to ethical reasons, since it is difficult to recruit patients with macular edema due to RVO and not treat them. In addition, it is not feasible to use healthy individuals as “controls” and subject them to treatment with intravitreal injections. Therefore, our results should be interpreted with caution, since one cannot be sure if the observed changes in retinal vessels’ diameters are attributed to the treatment with intravitreal injections or to the natural course of the disease, or both. We also did not measure ocular blood flow, taking into account that the vasoconstrictor effect of intravitreal anti-VEGF treatment on retinal vessels could lead to ocular blood flow disruption. It should be also mentioned that the basic working principle of our software is the measurement of the reflecting brightness derived from the erythrocyte column width. We acknowledge that this may result in an overestimation of the internal diameter [35]. Strengths of our study were that the software we used for static vessel analysis is validated [22], as well as the fact that all measurements were performed by one examiner. In addition, we included both CRVO and BRVO patients to generalize our results.

In conclusion, we found a significant vasoconstriction of both retinal arterioles and venules in patients with RVO 3 months after intravitreal ranibizumab treatment compared to baseline. This could be of clinical importance, since the degree of vasoconstriction might be an early marker of treatment efficacy, consistent with the idea that hypoxia is the major trigger of VEGF in RVO. However, our results may reflect a return to the normal diameter after disease-induced vasodilation, rather than true vasoconstriction. Future studies evaluating larger sample sizes are needed to confirm our findings and clarify the underlying pathophysiological mechanisms, leading to vasoconstriction of retinal vessels in RVO patients. Additionally, comparison between the impact of intravitreal anti-VEGF treatment and sustained-release dexamethasone implants would be of clinical importance.

## Figures and Tables

**Figure 1 jpm-13-00351-f001:**
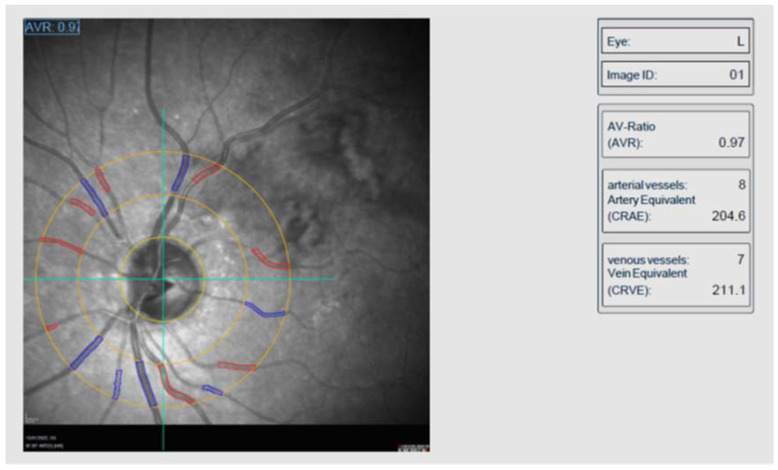
Retinal vessel calibers analysis using Vesselmap 2 software. Arterioles and venules are spread around the outer circle.

**Table 1 jpm-13-00351-t001:** Demographic and clinical characteristics of the study sample at baseline.

	N = 16 Patients/17 Eyes
Age (years, mean ± SD)	67 ± 10.2
Gender (N, %) *Male* *Female*	9 (56.3%) 7 (43.7%)
Retinal vein occlusion type (N, %) *Central* *Branch*	6 (37.5%) 10 (62.5%)
Best-corrected visual acuity (decimal, mean ± SD)	0.28 ± 0.13
Central subfield thickness (μm, mean ± SD)	427.4 ± 63.7

**Table 2 jpm-13-00351-t002:** Retinal vessel diameters before and after intravitreal ranibizumab treatment in RVO patients.

	Before	After	*p*-Value
**Retinal vein occlusion (n = 16)**			
**CRAE (μm)**	215.2 ± 11.2	201.2 ± 11.1	**<0.001**
**CRVE (μm)**	233.8 ± 29.6	207.6 ± 21.7	**<0.001**
**AVR**	0.93 ± 0.10	0.98 ± 0.12	0.107
**Branch retinal vein occlusion (n = 10)**			
**CRAE (μm)**	215.8 ± 10.3	203.6 ± 10.3	**0.002**
**CRVE (μm)**	221.0 ± 15.6	197.2 ± 12.2	**<0.001**
**AVR**	0.98 ± 0.07	1.04 ± 0.09	**0.043**
**Central retinal vein occlusion (n = 6)**			
**CRAE (μm)**	214.4 ± 14.2	197.5 ± 11.9	**0.007**
**CRVE (μm)**	253.5 ± 35.5	223.7 ± 23.7	**0.031**
**AVR**	0.86 ± 0.10	0.89 ± 0.10	0.434

CRAE: central retinal arteriolar equivalent; CRVE: central retinal venular equivalent; AVR: arteriolar to venular ratio.

## Data Availability

Data would be available upon request.

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
