# Peer review of "Alterations in Retinal Vessel Diameters in Patients with Retinal Vein Occlusion before and after Treatment with Intravitreal Ranibizumab"

_jpm, 2023, doi:10.3390/jpm13020351_

Round 1

Reviewer 1 Report

Congratulations to the authors for their work. Below are some of my specific questions.

- In Table 2, you summarized the different studies that have been done in this area before. Is there a feature that makes your work different from these?

- If we look at the literature information in Table 2, different results were obtained for Intravitreal Ranibizumab administration. You also mentioned the low number of subjects and the absence of a control group as the weaknesses of your study. Is the reliability of the software program alone sufficient to interpret your results as significant or not?

Author Response

“Congratulations to the authors for their work. Below are some of my specific questions.”

We would like to thank you for your comments and for the opportunity you gave us to revise our manuscript and present our work in a more elaborate way. Please find all changes within the manuscript, using track changes.

“In Table 2, you summarized the different studies that have been done in this area before. Is there a feature that makes your work different from these?”

We thank the reviewer for the comment. Our findings confirm the results of the existing literature, emphasizing the significance of measuring retinal vessel diameters in RVO patients with macular edema as a potential marker of treatment efficacy. Only two studies investigated the impact of intravitreal ranibizumab on retinal vessels. Our study differs from these two studies, since we included both CRVO and BRVO patients to generalize our results in RVO patients, while Sacu et al included only BRVO patients. Moreover, we had 3-month follow-up while previous study (Corvi et al) had 1-month follow-up. We added these comments in Discussion section (Discussion, Paragraph 2, lines 5-9). We deleted Table 2, as the 2nd reviewer suggested.

“If we look at the literature information in Table 2, different results were obtained for Intravitreal Ranibizumab administration. You also mentioned the low number of subjects and the absence of a control group as the weaknesses of your study. Is the reliability of the software program alone sufficient to interpret your results as significant or not?

We thank the reviewer for the comment. Our results were consistent with those of the previous studies using ranibizumab in RVO (Sacu et al and Corvi et al). Our measurements are based on a validated software, which is considered to be reliable and has been used in previous studies as well. The reliability of the program is not affected by the small sample size of the study.

Reviewer 2 Report

The manuscript by Evaggelia and collaborators is related to investigating the alterations of retinal vessel diameters in patients with macular edema secondary to retinal vein occlusion before and after treatment with intravitreal ranibizumab. However, a few points have to be completed. 

 1. It is essential to set a control group. We are curious to know if the decrease is caused by the injection, ranibizumab, or retinal vein occlusion so that everyone will question the conclusion.

2. What differentiates retina arterioles and venules around the outer circle? Is it auto-set by the machine? How does the machine recognize them? It directly affects the correctness of the results.

3. What are the criteria for the inclusion and exclusion of patients? It is essential to describe if the patient has systemic diseases. 

4. Table 2 looks unnecessary. It should be concluded patient information table.

Author Response

“The manuscript by Evaggelia and collaborators is related to investigating the alterations of retinal vessel diameters in patients with macular edema secondary to retinal vein occlusion before and after treatment with intravitreal ranibizumab. However, a few points have to be completed.”

We would like to thank you for your comments and for the opportunity you gave us to revise our manuscript and present our work in a more elaborate way. Please find all changes within the manuscript, using track changes.

  1. “It is essential to set a control group. We are curious to know if the decrease is caused by the injection, ranibizumab, or retinal vein occlusion so that everyone will question the conclusion.”

We would like to thank the reviewer for this valuable comment. In fact, this is a limitation of the study. However, no control group was recruited due to ethical reasons, as it is difficult to include patients with macular edema due to retinal vein occlusion and not treat them. Moreover, it is impossible to recruit healthy humans and apply them to intravitreal injections. To set a healthy control group with age and sex-matched individuals and compare them with patients with RVO and macular edema treated with intravitreal anti-VEGF agents could not answer the question if the reduction is attributed to the injection or the natural course of the disease or both. The above reasons could explain why we have not set a control group in the design of our study. We added these clarifications in Discussion section (Discussion, Paragraph 6, lines 2-8).

  1. “What differentiates retina arterioles and venules around the outer circle? Is it auto-set by the machine? How does the machine recognize them? It directly affects the correctness of the results.”

We thank the reviewer for the comment. Diameters of retinal arterioles and venules were manually measured by a very well trained/expert ophthalmologist (E.A.) blinded to clinical data. Retinal arterioles are smaller than retinal venules (1:2). Retinal venules are darker and more tortuous than arterioles. So, measurement was based on these criteria and was not affected by the machine. We have revised appropriately the section of materials and methods as follows: “All measurements were performed manually, based on the clinical knowledge that (a) retinal arterioles are smaller than venules and (b) retinal venules are darker and more tortuous than arterioles.” (Materials and methods, Paragraph 2, lines 15-17).

  1. “What are the criteria for the inclusion and exclusion of patients? It is essential to describe if the patient has systemic diseases.”

We would like to thank the reviewer for this comment. In fact, patients did not have any systemic diseases.  We added in Methods section the inclusion/exclusion criteria of patients, as follows:

“Inclusion criteria were the following: (1) patients aged older than 18 years and able to provide informed consent; (2) treatment naïve patients with macular edema secondary to RVO (central subfield thickness-CST ≥320 μm); (3) best-corrected visual acuity (BCVA) between 20/200 and 20/25 (Snellen equivalent) in the study eye. Patients were excluded from the study if they had diabetes mellitus, hypertension or other systemic disease, other retinal diseases than RVO, history of ocular inflammation, uncontrolled glaucoma with intraocular pressure (IOP) ≥30mmHg, previous laser photocoagulation, previous intravitreal injection of anti-VEGF or steroids, trauma, any ocular surgery within the previous 6 months and significant media opacities (from cornea or lens) that could preclude adequate retinal imaging, affecting the reliability of measurements.” (Materials and methods, Paragraph 1, lines 4-14).

  1. “Table 2 looks unnecessary. It should be concluded patient information table”.

We thank the reviewer for the comment. We have removed table 2 from the revised manuscript. We included a table with our patients’ data instead.

Reviewer 3 Report

1.      Abstract:

a.      Methods section can specify that the measurements were made before and "3 months" after the injection. Unable to understand Line 18. I think the review of the literature on this topic is an integral part of the discussion section, I don’t see the reason why it should be mentioned explicitly.

b.      Results can mention the overall values if possible.

2.      Introduction:

a.      Second sentence in Introduction – hard to compare 0.77% of global population in 2015 and 16 million today. It will be easy to compare if the percentage or number is given for both 2015 and today.

b.      Good work explaining the pathogenesis of RVO, etc, however, relevant to the paper would be a small paragraph summarizing what the other few studies that have examined the therapeutics effect have found. This will enable the readers to understand the need for the present study.

3.      Methods:

a.      Curious why people with HTN were excluded when HTN is the major risk factor for RVO. >30 mmHg as exclusion criteria seems like a lenient limit for IOP, did any of the subjects exceed 21 mmHg?

b.      How was the grader blinded to clinical data? What clinical data was blinded?

c.      The wordings on the calculations of CRAE, etc, give the impression that the authors did the calculations themselves, but if my understanding is right, these calculations are automatically done by the software and the values are printed, correct? Or it should be clarified whether the software was developed by the authors for the purpose of the study (as the author E.A. is given the credit for the software, clarity on this will be helpful).

4.      Figures:

a.      The sample retinal image of Figure 1 shows that the retinal vessels are not marked to their entire extent within the annulus. Some of the vessels are mislabeled as venules, especially in the temporal and inferior regions of the disc. This questions the accuracy of the measurements reported in the study.

5.      Tables:

a.      Suggestion: The vascular parameters can be removed from Table 1 because it is redundant. A separate table with values for BRVO and CRVO can be shown. This makes it easier to compare than reading the values from the text.

6.      Results:

a.      It would also be nice to see how the macular edema decreased after 3 months and see if there is any correlation between the amount of reduction in macular edema and changes in vascular measurements after treatment.

b.      How do these measurements of CRAE and CRVE at baseline and 3 months after injection compare with clinically normal population?

7.      Discussion:

a.      All of these previous studies, whether the treatment was anti-VEGF, laser, or steroids, show vasoconstriction for the most part (including the present study) except for a few. It would be beneficial to know the magnitude of vasoconstriction in each of these studies summarized in a table.

b.      Line 212- why are the results from the previous study controversial?

c.      Limitations of these measurements in the representation of the true caliber of the vessels can be discussed.

d.      A control group, i.e. clinically normal population without any intervention, can add value because it will give insight into normal variations in these measurements over time. As I understand, these vascular measurements vary based on many systemic factors such as cardiac cycle. And as the authors mention in the conclusion that these values “may reflect a return to the normal diameter”, this can be substantiated if they have normative values. Or their values can be compared with normative values available in the literature to substantiate it.

e.      I am not sure why a future study with a large sample size is required to confirm the findings from the present study as their findings are similar to many previous studies. Is there any specific part of the finding that the authors are doubtful about that they think a larger study is required to confirm it?

f.       A discussion on reduction in macular edema and its relation with vasoconstriction would be useful.

Author Response

Point-to-point response to reviewers’ comments

Reviewer 3

  1. Abstract:
  2. Methods section can specify that the measurements were made before and "3 months" after the injection. Unable to understand Line 18. I think the review of the literature on this topic is an integral part of the discussion section, I don’t see the reason why it should be mentioned explicitly.

Reply: We thank the reviewer for the comment. We have removed the sentence related to the review of the literature and we have modified as follows:"Digital retinal images were obtained from 16 patients and retinal vessel diameters were measured before and three months after intravitreal ranibizumab, with validated software to determine central retinal arteriolar and venular equivalents, as well as arteriolar to venular ratio". (Abstract, Methods, lines 2-3).

  1. Results can mention the overall values if possible.

Reply: We have taken into account reviewer's suggestion and modified appropriately, adding that: "Specifically, the Central Retinal Arteriolar Equivalent was 215.2±11.2μm at baseline and 201.2±11.1μm at month 3 after ranibizumab treatment (p<0.001), while the Central Retinal Venular Equivalent was 233.8±29.6 μm before treatment versus 207.6±21.7 μm at month 3 after ranibizumab treatment (p<0.001)”. (Abstract, Results, lines 4-7).

  1. Introduction:
  2. Second sentence in Introduction – hard to compare 0.77% of global population in 2015 and 16 million today. It will be easy to compare if the percentage or number is given for both 2015 and today.

Reply: We thank the reviewer for the comment. We have corrected as follows: "Rogers et al tried for the first time to estimate the prevalence of RVO worldwide and found that 16.4 million people aged 30 years and above were suffering from RVO in 2008. In 2015, the global prevalence of RVO was found to be about 0.77% in people aged 30-89, affecting about 28 million people worldwide.” (Introduction, Paragraph 1, lines 3-7).

 Good work explaining the pathogenesis of RVO, etc, however, relevant to the paper would be a small paragraph summarizing what the other few studies that have examined the therapeutics effect have found. This will enable the readers to understand the need for the present study.

Reply: We thank the reviewer for the comment. We have mentioned the results of the few studies we found in the literature extensively in the discussion section. However, we added a summary of the findings of previous studies in introduction, according to your suggestion, as follows: “The majority of previous studies reported that there was vasoconstriction of arterioles and venules after intravitreal anti-VEGF or laser treatment in RVO patients. However, the so far studies had short-term follow-up of 1-month, while they included only patients with BRVO.”  (Introduction, Paragraph 4, lines 4-7).

  1. Methods:
  2. Curious why people with HTN were excluded when HTN is the major risk factor for RVO. >30 mmHg as exclusion criteria seems like a lenient limit for IOP, did any of the subjects exceed 21 mmHg?

Reply: Hypertension has a strong vasoconstrictor impact on retinal vessel diameters, mainly on arterioles and would act as a confounder in our analysis. Our purpose was to exclude patients with any condition that according to literature affects retinal vessel calibers, so as to make our results more reliable.

Glaucoma is also an important risk factor for RVO. The few other previous studies have also used the same exclusion criteria. In order to compare our findings with those of other studies and eliminate possible factors of heterogeneity, we have designed appropriately our study.

No patient had IOP >21 mmHg. The mean IOP at baseline in our study was 18.3±2.1 mmHg. We added this comment in Results section (Results, Paragraph 1, line 5).

 How was the grader blinded to clinical data? What clinical data was blinded?

Reply: We thank the reviewer for the comment. The grader did not know any information about patient's history or OCT image and did not know if the photo that measured was from a patient before or after treatment. We added this clarification in the manuscript (Methods, Paragraph 2, lines 11-12).

  1. The wordings on the calculations of CRAE, etc, give the impression that the authors did the calculations themselves, but if my understanding is right, these calculations are automatically done by the software and the values are printed, correct? Or it should be clarified whether the software was developed by the authors for the purpose of the study (as the author E.A. is given the credit for the software, clarity on this will be helpful).

Reply: We thank the reviewer for the comment. It is clearly written in the manuscript that our software is validated and measurements were done manually. We cite the related sentences: "For each photograph, the calibers of the six largest retinal arterioles and venules passing through a zone between 0.5 and 1.0 disc diameters from the optic disc margin were measured and analyzed using a Static Retinal Vessel Analyzer (SVA-T and Vesselmap 2 software [22], Visualis, Imedos Systems UG, Jena, Germany) (Figure 1). All measurements were performed manually, based on the clinical knowledge that (a) retinal arterioles are smaller than venules and (b) retinal venules are darker and more tortuous than arterioles.”

However, we added a phrase in the manuscript to be clearer for the reader “Then, the calculations were automatically done by the validated software.”(Methods, Paragraph 2, line 19).

  1. Figures:
  2. The sample retinal image of Figure 1 shows that the retinal vessels are not marked to their entire extent within the annulus. Some of the vessels are mislabeled as venules, especially in the temporal and inferior regions of the disc. This questions the accuracy of the measurements reported in the study.

Reply: We thank the reviewer for the comment. It is not necessary for the accuracy of the measurements to mark the whole extent of the vessel, it is sufficient to mark the outer segment of the vessel near the outer margin of the zone. These instructions are given by the manufacturer of the software. The vessels are not mislabeled, we have explained how retinal arterioles were distinguished from retinal venules (Methods, Paragraph 2, lines 12-19). It is already mentioned in our manuscript that "The intra-observer reproducibility of retinal vascular measurements was excellent as indicated by the intraclass correlation coefficient (>0.9)".

  1. Tables:
  2. Suggestion: The vascular parameters can be removed from Table 1 because it is redundant. A separate table with values for BRVO and CRVO can be shown. This makes it easier to compare than reading the values from the text.

Reply: We thank the reviewer for the comment. We have removed values of CRAE, CRVE and AVR from table 1, and we provided data for both BRVO and CRVO on Table 2.

  1. Results:
  2. It would also be nice to see how the macular edema decreased after 3 months and see if there is any correlation between the amount of reduction in macular edema and changes in vascular measurements after treatment.

We added in the manuscript that “The mean CST was 427.4±63.7 μm at baseline and significantly decreased to 387.2±45.8 μm at month 3 after intravitreal ranibizumab treatment” (Results, Paragraph 1, lines 6-8).

No correlation between the reduction in macular edema and changes in vascular measurement were found. We added this comment in the manuscript (Results, Paragraph 1, lines 13-15).

 How do these measurements of CRAE and CRVE at baseline and 3 months after injection compare with clinically normal population?

Reply: We thank the reviewer for the comment. There are not normal values for CRAE and CRVE, except from AVR. However, AVR carries information for both retinal arteriolar and venular diameter, so is not useful for clinical interpretation.

  1. Discussion:
  2. All of these previous studies, whether the treatment was anti-VEGF, laser, or steroids, show vasoconstriction for the most part (including the present study) except for a few. It would be beneficial to know the magnitude of vasoconstriction in each of these studies summarized in a table.

Reply: We thank the reviewer for the comment. This could not be summarized in a table, as it is not reported in these studies, which evaluate different therapeutic procedures. It is only written if the differences before and after treatment are statistically significant. Furthermore, even in the three previous studies where anti-VEGF were used, the two studies used the same with us software to determine retinal vessel calibers. In conclusion, due to the heterogeneity of the findings, the software used, as well as of the different therapeutic procedures that were applied and the fact that there are not normal values in the literature for CRAE and CRVE, the magnitude of vasoconstriction could not be depicted.

  1. Line 212- why are the results from the previous study controversial?

Reply: We thank the reviewer for the comment. The results are controversial as far as the findings of retinal venular diameters are concerned in patients with BRVO and CRVO between studies conducted by Eibenberger and Yilmaz Tugan et al.

  1. Limitations of these measurements in the representation of the true caliber of the vessels can be discussed.

Reply: We thank the reviewer for the comment. We have added the following sentence in the limitations section of the discussion. "The basic working principle of our software is the measurement of the reflecting brightness derived from the erythrocyte column width. We acknowledge that this may result in an overestimation of the internal diameter." (Discussion, Paragraph 6, lines 12-15).

  1. A control group, i.e. clinically normal population without any intervention, can add value because it will give insight into normal variations in these measurements over time. As I understand, these vascular measurements vary based on many systemic factors such as cardiac cycle. And as the authors mention in the conclusion that these values “may reflect a return to the normal diameter”, this can be substantiated if they have normative values. Or their values can be compared with normative values available in the literature to substantiate it.

Reply: We thank the reviewer for the comment. As already mentioned, there are not normal values for CRAE and CRVE, except from AVR. However, AVR carries information for both retinal arteriolar and venular diameter, so is not useful for clinical interpretation. This was also another reason that no control group was recruited in the design of our study. In addition, no control group was included due to ethical reasons, since it is difficult to recruit patients with macular edema due to RVO and not to treat them. In addition, it is not feasible to use healthy individuals as “controls” and subject them to treatment with intravitreal injections. Therefore, our results should be interpreted with caution, since one cannot be sure if the observed changes in retinal vessels’ diameters are attributed to the treatment with intravitreal injections or to the natural course of the disease or both. We have discussed this as limitation of the study (Discussion, Paragraph 6).

  1. I am not sure why a future study with a large sample size is required to confirm the findings from the present study as their findings are similar to many previous studies. Is there any specific part of the finding that the authors are doubtful about that they think a larger study is required to confirm it?

Reply: We thank the reviewer for the comment. The small sample size is mentioned in the limitations of our study. In fact, we would like to compare therapeutic alternatives regarding the changes in vasoconstriction of retinal vessels. Therefore, we have added the following section in the last paragraph of our discussion. "Comparison between the impact of anti-VEGF treatment and sustained-release intravitreal dexamethasone implant in the same study would be of clinical importance" (Discussion, Paragraph 7, lines 9-11).

  1. A discussion on reduction in macular edema and its relation with vasoconstriction would be useful.

Since no correlation was found between the reduction in macular edema and changes in the vessels, we did not discuss this finding in the manuscript.

Round 2

Reviewer 1 Report

Thanks to the authors for all their responses.

Author Response

Thank you for your comments and for the opportunity to publish our study. 

Reviewer 2 Report

Is the information shown in the table1 for the preliminary or treatment data?

Fewer patients enrolled, and all measurements were performed manually, also lacking the control group (whether the SD-OCT change is from the injection, the ranibizumab, or the retinal vein occlusion?). The results are questionable.

Author Response

“Is the information shown in the table1 for the preliminary or treatment data?”

We would like to thank you for your comments and for the opportunity you gave us to revise our manuscript and present our work in a more elaborate way. Please find all changes within the manuscript, using track changes.

Table 1 refers to baseline data. We added this clarification in the manuscript (Results, Paragraph 1, line 2 and in the legend of Table 1).

 “Fewer patients enrolled, and all measurements were performed manually, also lacking the control group (whether the SD-OCT change is from the injection, the ranibizumab, or the retinal vein occlusion?). The results are questionable.”

Thank you for your comment.

The small sample size and the lack of control group are limitations of the study. However, it is not ethical to include a control group, which will not receive treatment for macular edema due to RVO. This is an inherent limitation in all previous studies as well, with the same design due to ethical reasons. That’s why we also agree that the results of such studies should be interpreted with caution. We have already analyzed this in the discussion section (Discussion, Paragraph 6, lines 3-9).

Nevertheless, our study has used a validated software, so as to provide accurate data for vessel analysis and all measurements were performed by the same qualified grader to avoid bias. In addition, most of previous studies provide 1-month results, we present 3-months results using ranibizumab for RVO.

Round 3

Reviewer 2 Report

Modifications have greatly improved the article, but I am still questioning the rigor of the results.